# Intestinal endometriotic nodules with a length greater than 2.25 cm and affecting more than 27% of the circumference are more likely to undergo segmental resection, rather than linear nodulectomy

Helizabet Abdalla-Ribeiro[1,2], Marina Miyuki Maekawa[1]*, Raquel Ferreira Lima[1], Ana Luisa Alencar de Nicola[1], Francisco Cesar Martins Rodrigues[1], Paulo Ayroza Ribeiro[1,2]

1 Department of Obstetrics and Gynecology of Santa Casa de de Misericórdia São Paulo, Sector of Gynecological Endoscopy and Endometriosis at Santa Casa de São Paulo, São Paulo, São Paulo, Brazil, 2 School of Medical Science of Santa Casa de Misericórdia de São Paulo, São Paulo, São Paulo, Brazil

☉ These authors contributed equally to this work.
* dramarinamaekawa@gmail.com

## Abstract

### Study objective

To analyze the efficacy of intestinal ultrasonography with bowel preparation (TVUSBP) for endometriosis mapping in evaluating intestinal endometriosis to choose the surgical technique (segmental resection or linear nodulectomy) for treatment.

### Design

Cross-sectional observational study.

### Setting

University Hospital—Center for Advanced Endoscopic Gynecologic Surgery from April 2010 to November 2014.

### Patient(s)

One hundred and eleven women with clinically suspected endometriosis and intestinal endometriotic nodule or intestinal adherence in TVUSBP for endometriosis mapping.

### Intervention(s)

All patients with suspected endometriosis underwent TVUSBP for endometriosis mapping prior to videolaparoscopy for complete excision of endometriosis foci, including intestinal foci, using the linear nodulectomy or segmental resection techniques, depending on the characteristics of the intestinal lesion with confirmation of endometriosis on anatomopathological examination.

**Data Availability Statement:** All relevant data are within the paper.

**Funding:** The author(s) received no specific funding for this work.

**Competing interests:** The authors have declared that no competing interests exist.

## Measurements and main results

Preoperative ultrasonographic assessment of the length of the intestinal nodule, circumference of the intestinal loop affected by the endometriotic lesion, distance from the anal verge and intestinal wall layers infiltrated by endometriosis, as well as other endometriosis sites. Of the 111 patients who participated in the study, 63 (56.7%) presented intestinal endometriotic nodules in ultrasonography, performed by a single examiner (A.L.A.N.), and underwent intestinal surgical treatment of deep endometriosis—linear nodulectomy or segmental resection. The analysis of the receiver operating characteristic (ROC) curve showed that a longitudinal length of the intestinal nodule of 2.25 cm and a loop circumference of 27% are cutoff points separating linear nodulectomy from segmental resection techniques for excising intestinal endometriosis. The information obtained by TVUSBP helps the surgeon and patient, in the preoperative period, to select the surgical technique to be performed for resection of intestinal endometriosis and plan the surgical procedure while taking into account postoperative morbidity.

## Introduction

The prevalence of intestinal involvement is estimated at 45–56% of patients with deep infiltrating endometriosis [1, 2].

Studies with questionnaires applied to patients with intestinal deep infiltrating endometriosis show that there is an 85–95% improvement in quality of life after surgery [3–7]. Given this improvement, when we opt for surgical treatment of intestinal endometriosis, different surgical techniques are available depending on the characteristics of the intestinal nodule, such as longitudinal length, circumference of the affected intestinal loop, depth and distance from the anal verge [8]. Among surgical techniques for intestinal endometriosis, one can choose nodulectomy by "shaving" [8–10], "mucosal skinning" [10], discoid resection [11–16], linear nodulectomy [17–22] or segmental resection [17, 20, 23, 24].

Although the final decision on which surgical technique is to be used, is always established during the intraoperative period; some studies suggest that the characteristics of the intestinal nodule (longitudinal length, circumference of the intestinal loop affected, layer affected and distance from the anal verge) obtained from a preoperative TVUSBP can help the surgeon decide which intestinal endometriosis resection technique is more likely to be performed [25].

Regarding the longitudinal length of the intestinal nodule, there is a tendency to perform segmental resection in single nodules larger than 3 cm [9, 13, 25]. Nodulectomy of larger nodules may lead to stenosis of the stapling area; for this reason, some authors suggest that intestinal nodules that infiltrate more than the internal muscular layer or when it affects more than 40% of the circumference of the loop, it should be subjected to segmental resection [13]. Regarding the distance from the anal verge of the intestinal endometriotic nodule, it is suggested that in excision cases of low intestinal lesions—defined as lesions located less than 5–8 cm from the anal verge—the rate of postoperative complication, with dehiscence or fistula or low anterior rectal resection syndrome, increases from 3–7% to 20% [14, 15, 17].

Evaluating which surgical technique is more likely to be used before surgery, accordingly to TVUSBP, is useful in preoperative planning and advising the patient about morbidity and complications related to each technique [8, 18].

The focus of this study is to provide information using preoperative TVUSBP that helps the surgeon in choosing the surgical technique to be used in the treatment of intestinal endometriosis: linear nodulectomy or segmental resection.

## Methods

### Study design

The study was approved by the ethics committee of the institution (Comitê de Ética em Pesquisa em Seres Humanos da Santa Casa de Misericórdia de São Paulo) under protocol number CAAE 59860916.1.0000.5479, with authorization to review medical records and videos of surgeries.

The consent form for this specific study was not obtained because in our institution an informed consent is not necessary for retrospective studies without evaluation, contact, questionnaire or interviews with the patients. In addition once interned at the institution, patients provide written consent to have data of their medical records used in research. All data was fully anonymized before the authors accessed them. One year after the surgical procedure, patients had no clinical follow up at the institution and continue with routine care at primary health care units.

In 2019, a cross-sectional observational study was performed and submitted to statistical analysis. This study included 111 medical records of patients diagnosed with endometriosis who underwent videolaparoscopy surgery from April 2010 to November 2014. These patients had clinical symptoms, complaints of infertility or physical examinations that would suggest endometriosis. They were submitted to ultrasonography, performed by a single examiner, suggesting intestinal endometriotic nodules—or images that the examiner can presume intestinal adhesion is present. All patients underwent videolaparoscopy surgery for intestinal endometriosis resection with endometriosis confirmation in anatomopathological examination. The surgeons were aware of the USTVBP result.

We included in this study all patients submitted to surgical treatment of bowel endometriosis in our department, by the same surgical team (P.A.A.R., H.S.A.A.R. and F.C.M.R.), during the specified period (2010–2014) regardless of age, parity, previous hormonal or surgical treatment for endometriosis or associated procedures. All patients had a TVUSBP examination performed by a single radiologist (A.L.A.N.) and their surgeries were recorded on DVD. Histological confirmation of intestinal endometriosis was a mandatory inclusion criteria.

The exclusion criteria for the study included: loss of medical records, preoperative diagnosis of endometriosis by another image exam, preoperative TVUSBP performed by another radiologist, intestinal resection not performed, intestinal resection done without the use of stapling; surgeries performed by a different surgical team.

Of the 111 medical records, 63 patients met all inclusion criteria and were divided into two groups: 36 undergoing segmental resection and 27 undergoing linear nodulectomy (Fig 1).

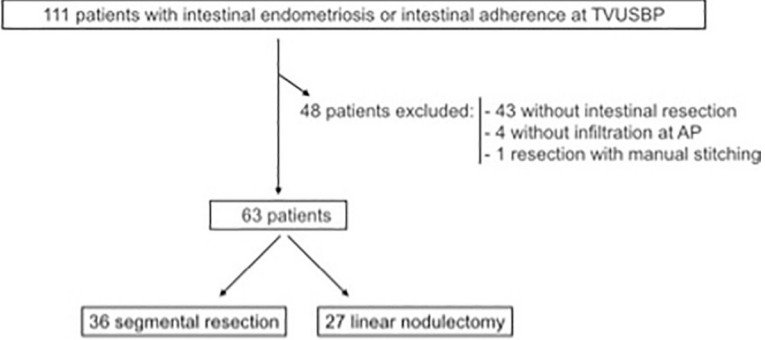

**Fig 1. Flowchart of patient selection for the study.**

There were 43 patients excluded for not having intestinal endometriosis infiltration at laparoscopy, 4 patients were excluded for not having intestinal endometriosis confirmed in AP examination and 1 patient was excluded because the surgical technique didn´t use staplers, which might have led to different complications.

The variables evaluated were age, preoperative symptoms, duration of symptoms, duration of previous treatment, previous surgeries for endometriosis and intra and postoperative complications.

Ultrasonography assessed the following parameters: longitudinal length of the intestinal nodule, circumference of the intestinal loop affected, distance from the anal verge and layer affected by the intestinal nodule. In addition, the presence of ovarian endometrioma, round ligament, bladder, vaginal, ureter, retrocervical and uterosacral ligament endometriosis was assessed [26].

The surgical data was evaluated by reviewing videos stored on DVD and classified according to the American Fertility Society (AFSr) criteria; the surgical time was calculated (from introduction to removal of the optic device from the abdominal cavity), and intraoperative complications were evaluated.

## Ultrasonography scanning technique

Transvaginal ultrasonography with bowel preparation for endometriosis mapping (USTVBP) was performed according to the protocol of our department [19] and literature [23, 27, 28]. All examinations were performed by the same examiner (A.L.A.N.). The ultrasound devices used were the GE Voluson S6 (GE Healthcare, Zipf, Austria) or IU 22 (Phillips Healthcare, Eindhoven, Netherlands) with 5-9-MHz transducers.

Intestinal deep infiltrating endometriosis lesions were defined as hypoechoic nodular thickening with regular or toothed margins (comet shape) or hypoechoic linear thickening with regular or irregular margins and involvement of the muscular or submucosa layers [26, 28] (Figs 2–4).

## Surgical procedures

The surgeries were performed by senior surgeons with extensive experience in the treatment of deep infiltrating endometriosis (P.A.A.R., H.S.A.A.R. and F.C.M.R.). The surgeons were

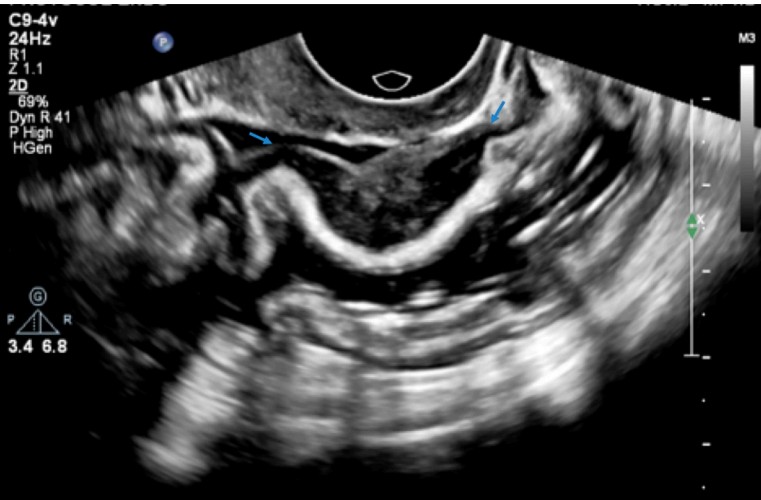

**Fig 2. USTVBP image showing a longitudinal section of the sigmoid.** Deep endometriosis nodule in the anterior wall with involvement up to the inner muscle layer (blue arrows).

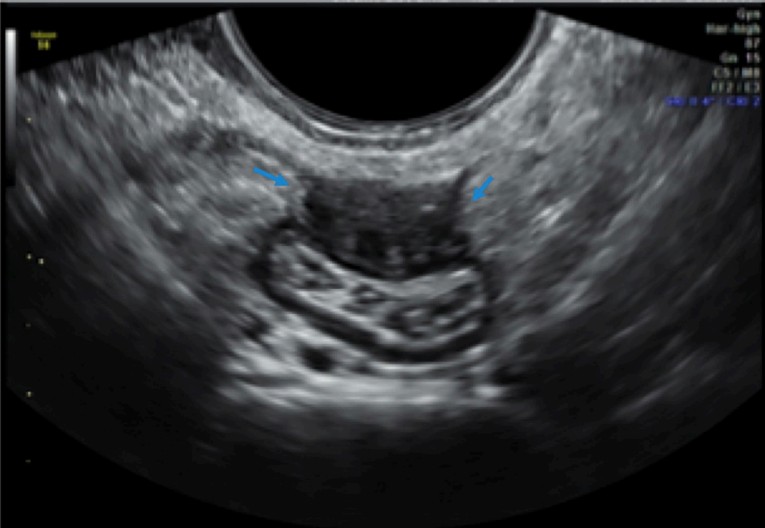

**Fig 3. USTVBP image showing an axial section of the rectosigmoid segment.** Deep endometriosis nodule in the anterior wall with circumferential involvement of about 30% (blue arrows).

aware of the USTVBP result. All surgeries were performed laparoscopically using a high-definition (HD) camera and a Xenon Nova 300 W light source, both from Storz. Access to the abdominal cavity was obtained using the closed technique with a Veress needle, incision of the umbilical scar and subsequent umbilical puncture with an 11-mm trocar. Three accessory punctures were performed with 5-mm trocars in the usual triangular arrangement. $CO_2$ was used to distend the cavity, and the surgeries were recorded on DVD. The surgeries followed the standard procedure of our institution [20], including the dissection of the retroperitoneal spaces, isolation of the ureters and nerve preservation. During the surgical procedure, harmonic energy and bipolar energy were used for dissection and coagulation as needed. All extra intestinal foci of endometriosis was removed prior to treatment of the intestinal nodule.

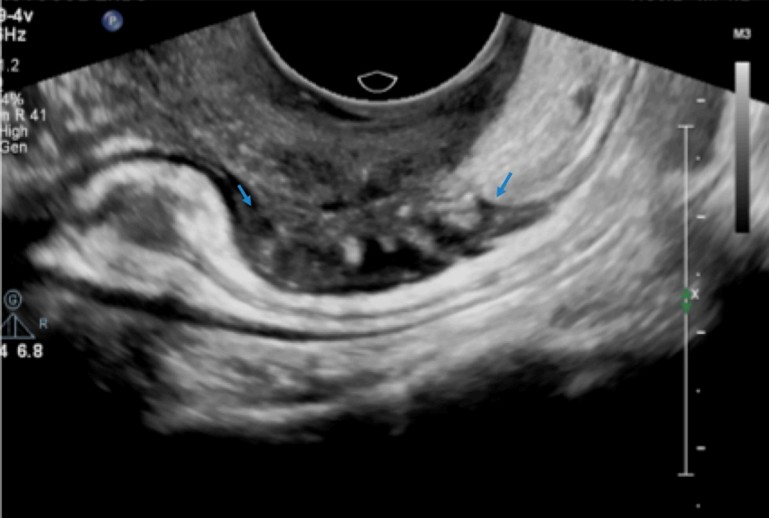

**Fig 4. USTVBP image showing a longitudinal section of the rectosigmoid segment.** Deep endometriosis nodule in the anterior wall with involvement up to the outer muscular layer (blue arrows).

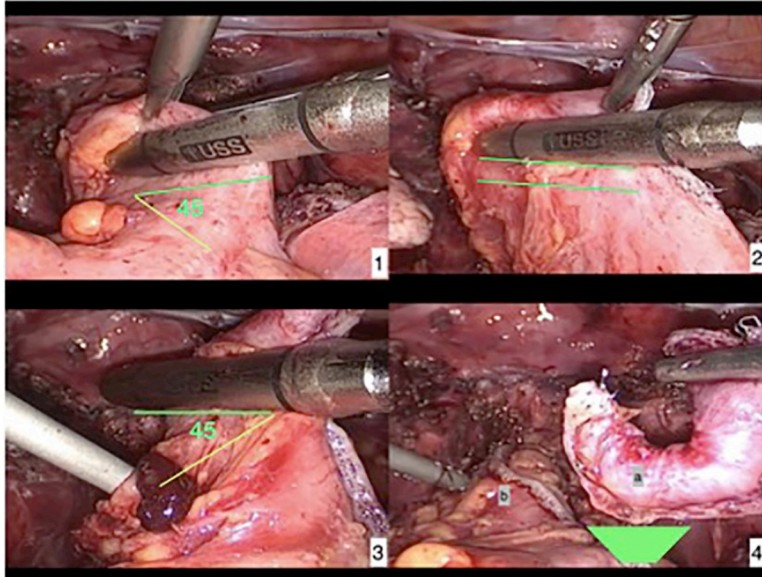

**Fig 5. Surgical steps of the linear nodulectomy technique.** 1- In the first step, the linear stapler was positioned 45 degrees from the axis of the intestinal lumen. 2- Next, consecutive staples (usually 2 or 3) were placed below the lesion and parallel to the axis of intestinal lumen. 3- To finish resection, the last stapler was positioned 45 degrees from the axis of the intestinal lumen to completely remove the endometriosis nodule. 4a- final aspect of the surgical specimen (inverted trapezoid shape). 4b- final aspect of the stapling line with a 29mm probe rectally inserted.

In the linear nodulectomy technique, the central portion of endometriosis in which there is infiltration of the intestinal wall was isolated using the serous layer shaving technique [16, 20]. This allowed the isolation of the point with intestinal infiltration, and reduced the removed area of the healthy intestinal wall. A locking forceps was used for traction of the intestinal nodule, and a linear stapler was introduced through the trocar located in the right iliac fossa; the linear stapler was positioned below the lesion (Fig 5). A 29-mm probe was inserted rectally to assess and confirm that there was no intestinal lumen stenosis, before and/or after stapling [20–22].

In the segmental resection technique, transverse linear stapling was performed caudally to the intestinal nodule. Next, the rectosigmoid segment was externalized through an incision in the right iliac fossa for resection under direct view. The proximal margin of the rectosigmoid was then prepared for anastomosis with a circular stapler (CDH33, Ethicon-Brazil). The incision was closed, and a 12-mm trocar was inserted so that the intestinal anastomosis procedure could be finalized by laparoscopy [20, 24].

Although the USTVBP can preoperatively define the dimensions of the lesions and suggest one technique or another, the final decision on the surgical technique to be performed (linear nodulectomy or segmental resection) was stablished, intraoperatively, by performing a rectal lumen diameter test, with the insertion of a 29 mm diameter rectal probe. If lumen stenosis was observed, the segmental resection technique was chosen.

## Statistic method

To determine the sample size, a pilot study was conducted with 10 patients, evaluating the length of the intestinal nodule in preoperative TVUSBP. This data was used to detect a statistically significant difference between the two groups at a significance level of 5% (alpha error) and a test power of 99.9%. The calculated sample size for each group (linear nodulectomy and

segmental resection) was 27 cases, with a median length of 1.8 cm and SD of 0.8. The difference in the mean intestinal nodule length between the two techniques was 2.9 cm.

Fisher's exact test was used to evaluate the qualitative variables. To test the normality of quantitative samples, we used the Kolmogorov-Smirnov test or the Shapiro-Wilk test.

Variables followed a non normal distribution and are expressed as the median and minimum and maximum variation (range). The Mann-Whitney test was used to evaluate the correlations between numerical variables and categorical variables. The Kruskal-Wallis test was used to compare quantitative variables from three or more groups of data. The chi-square test was used to evaluate the associations between categorical variables, i.e., all qualitative variables, including the ordinal variables. For correlations between numerical variables, Spearman's correlation coefficient was applied. For inferential analyses, a significance level ($\alpha$) of 5% was adopted.

ROC curves were constructed to determine the cutoff points of the following variables: diameter of the intestinal nodule, circumference of the loop affected and distance from the anal verge.

## Results

Of the 111 records analyzed, 63 met all inclusion and exclusion criteria. Of these 63 patients, 27 underwent linear nodulectomy, and 36 underwent segmental resection.

The median age of the patients was 37 years with a range of 27–51 years for the nodular nodulectomy technique and 34 years with a range of 28–46 years for segmental resection; there was no significant difference between the techniques performed (Table 1).

The median surgical time was 90 minutes with a range of 35–180 minutes for the group undergoing linear nodulectomy and 120 minutes with a range of 60–240 minutes for the segmental resection group. The p-value of 0.005 indicated a significant difference (Table 1).

The median duration of symptoms before surgery was 36 months with a range of 1–240 months for linear nodulectomy and 48 months with a range of 12–288 months for surgical resection, with no evidence of a significant difference between the groups (Table 1).

Of the patients undergoing linear nodulectomy, only 7.4% were asymptomatic, but complained about infertility, and 81.4% had at least one of the symptoms of pelvic/lumbar pain, such as dysmenorrhea, dyspareunia, low back pain or chronic pelvic pain. Approximately 3.7% of patients had hematochezia, and 7.4% had menorrhagia or metrorrhagia, as shown in Fig 6.

**Table 1. Demographic and surgical data of 63 patients that underwent intestinal endometriosis resection by linear nodulectomy or segmental resection at Santa Casa de São Paulo, 2010–2014.**

| Characteristic | Linear nodulectomy median (Range) | Segmental resection median (Range) | P- value |
|---|---|---|---|
| Age (years) | 37 (27–51) | 34 (28–46) | 0,284 |
| Duration of symptoms (months) | 36 (1–240) | 48 (12–288) | 0,362 |
| Duration of previous treatment (months) | 24 (7–120) | 36 (6–216) | 0,419 |
| Surgical time (min) | 90 (35–180) | 120 (60–240) | 0, 005 |
| Intraoperative complications (%) | 0 | 0 | Not applicable |
| Postoperative complications (%) | 2 (7.4) | 4 (11,1) | 0,951 |
| Extrapelvic endometriosis (%) | 3,(11,1) | 13 (36,1) | 0, 050 |
| Prior drug treatment (%) | 18 (66,6) | 27 (75) | 0,770 |
| Previous surgery for endometriosis (%) | 12 (44,4) | 14 (38,8) | 0,853 |

Note: The data is expressed as the median (range) (if non normal distribution) or n (percentage).

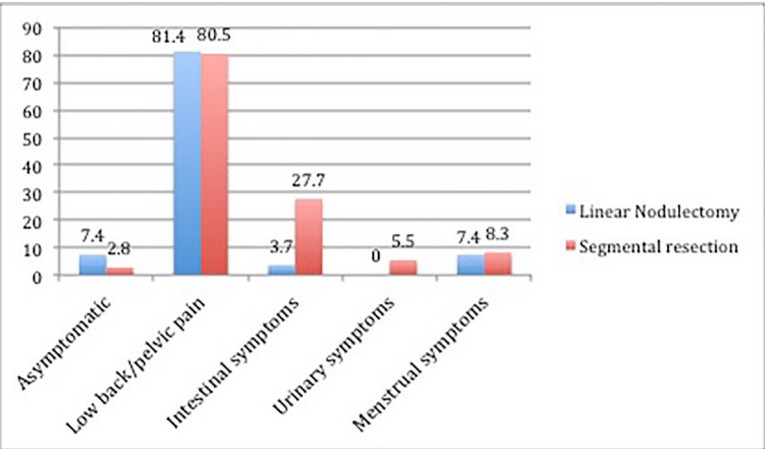

**Fig 6. Relationships of the linear nodulectomy and segmental resection techniques with symptoms prior to surgery in patients with intestinal deep infiltrating endometriosis at Santa Casa de São Paulo, 2019.** The data is expressed in percentages.

In patients undergoing segmental resection, 2.8% were asymptomatic. Approximately 80.5% had at least one symptom of pelvic/lumbar pain, such as dysmenorrhea, dyspareunia, low back pain or chronic pelvic pain. Regarding intestinal symptoms, 27.7% had at least one intestinal symptom, such as anal pain, dyschezia, diarrhea, pencil-thin stools, constipation, flatulence or hematochezia. Only 5.5% of the patients had symptoms of dysuria or recurrent urinary tract infection, and 8.3% had complaints of menorrhagia or metrorrhagia, as shown in Fig 6. We did not find a significant difference between the techniques with respect to symptoms.

Regarding the presence of extraintestinal endometriosis, we found greater involvement in patients undergoing intestinal endometriosis excision with the segmental resection technique (36,1%) compared to the linear nodulectomy technique (11.1%), with statistical difference (p = 0.050). Patients submitted to the first technique, showed involvement of the appendix, sigmoid, descending colon, caecum, ileum, left uterine artery and abdominal wall. Among the patients undergoing linear nodulectomy, there was involvement of the left uterine artery, piriformis muscle, left pudendal nerve, left sciatic nerve and right hypogastric nerve.

The extent of the disease was assessed using the AFSr criteria while reviewing the surgery videos. Severe endometriosis (stage IV) was present in 69.8% of cases; however, there was no significant difference between the resection techniques performed.

We analyzed the presence of endometriosis in other extraintestinal locations (ovaries, round ligament, bladder, vagina, ureter, retrocervical area and uterosacral ligaments) related to the intestinal surgical technique; however, we found no significant difference.

The analysis of the receiver operating characteristic (ROC) curve showed a cutoff point of 10.5 cm for the distance from the anal verge. Values below this cutoff point were associated with the segmental resection technique, while values above it were associated with linear nodulectomy. Sensitivity and specificity were calculated for each distance from the anal verge, and for this cutoff value, we found a sensitivity of 76.7% and specificity of 53.6% (Table 2) (Fig 7). A positive predictive value (PPV) of 63.9% and a negative predictive value (NPV) of 68.2% were found. The median for the distance from the anal verge for the nodulectomy group was 11.25cm (Range 8–19) and, for the segmental resection group, it was 10cm (Range 6–17), with p value 0.033. As additional data, there was a significant difference in the relationship of the distance of the endometriosis nodule from the anal verge and with the presence of vaginal

**Table 2. ROC curve parameters according to sensitivity and specificity values.**

| Cut off values of distance from anal verge (cm) | Sensitivity | 1-Specificity | Cut off values of length (cm) | Sensitivity | 1-Specificity | Cut off values of circumference | Sensitivity | 1-Specificity |
|---|---|---|---|---|---|---|---|---|
| 5 | 1 | 0 | 0,35 | 1 | 0,04 | 11,00% | 1 | 0 |
| 6,5 | 1 | 0,071 | 0,85 | 1 | 0,2 | 13,50% | 0,977 | 0,105 |
| 7,5 | 1 | 0,214 | 1,1 | 0,972 | 0,44 | 17,50% | 0,977 | 0,316 |
| 8,5 | 0,967 | 0,214 | 1,25 | 0,972 | 0,52 | 19,50% | 0,953 | 0,526 |
| 9,5 | 0,867 | 0,357 | 1,55 | 0,944 | 0,6 | 22,50% | 0,884 | 0,526 |
| 10,5 | 0,767 | 0,536 | 1,85 | 0,917 | 0,72 | 25,50% | 0,721 | 0,737 |
| 11,25 | 0,5 | 0,643 | 2,15 | 0,889 | 0,84 | 27,00% | 0,721 | 0,789 |
| 11,75 | 0,467 | 0,643 | 2,25 | 0,889 | 0,92 | 29,00% | 0,674 | 0,789 |
| 12,5 | 0,333 | 0,821 | 3,1 | 0,694 | 0,96 | 34,00% | 0,558 | 0,895 |
| 13,5 | 0,267 | 0,821 | 3,55 | 0,611 | 1 | 37,50% | 0,419 | 0,947 |
| 14,5 | 0,233 | 0,893 | 4,15 | 0,417 | 1 | 39,00% | 0,395 | 0,947 |
| 15,5 | 0,133 | 0,929 | 4,9 | 0,278 | 1 | 42,00% | 0,233 | 1 |
| 16,5 | 0,1 | 0,964 | 5,45 | 0,222 | 1 | 49,00% | 0,116 | 1 |
| 17,5 | 0,067 | 1 | 6,85 | 0,139 | 1 | 55,00% | 0,047 | 1 |
| 18,5 | 0,033 | 1 | 9,05 | 0,056 | 1 | 65,00% | 0,023 | 1 |
| 20 | 0 | 1 | 18 | 0 | 1 | 71,00% | 0 | 1 |

endometriotic nodules, with greater distances from the anal verge for intestinal nodules that did not have associated vaginal nodules (median of 12.9 cm) and smaller distances (median of 8.4 cm) when the vagina was involved, with p = 0.001.

For the longitudinal length of the intestinal nodule, the ROC curve showed that a value of 2.25 cm was the best equilibrium point between sensitivity (88.9%) and specificity (92%) (Table 2) (Fig 7). A positive predictive value (PPV) of 94.1% and a negative predictive value (NPV) of 85.1% were found. Thus, linear nodulectomy would be used for nodules smaller than 2.25 cm, and segmental resection would be used for nodules larger than 2.25 cm. The median for the longitudinal length of the intestinal nodule for the nodulectomy group was 1.2cm (range 0.3–3.5) and, for the segmental resection group, it was 3.9cm (range 1–17), with p value <0.001. When evaluating the presence of endometriotic nodules in the vagina, there was a significant difference, with smaller intestinal nodule diameters (median of 2.3 cm) in cases in which there was no vaginal involvement and larger diameters (median of 4.9 cm) when there was endometriosis of the vagina, with p = 0.019.

Application of ROC curve analysis to the percentage of circumference of the loop affected by the intestinal endometriotic nodule identified a value of 27% as the best equilibrium point between sensitivity (72.1%) and specificity (78.9%) (Table 2) (Fig 7). A positive predictive value (PPV) of 88% was found as well as a negative predictive value (NVP) of 55%. Values higher than this cutoff point of 27%, were associated with the segmental resection technique, while lower values were associated with the linear nodulectomy technique. The median for the percentage of circumference of the loop affected by the intestinal endometriotic nodule for the nodulectomy group was 19% (range 12–40) and, for the segmental resection group, it was 35% (range 12–70), with p value <0.001.

Regarding the affected intestinal layer, in the segmental resection technique, 69.4% of cases had mucosal and submucosal involvement, and in the linear resection technique, 93.1% of cases exhibited involvement of the muscular and serosa layers, with p<0.001 showing statistical significance.

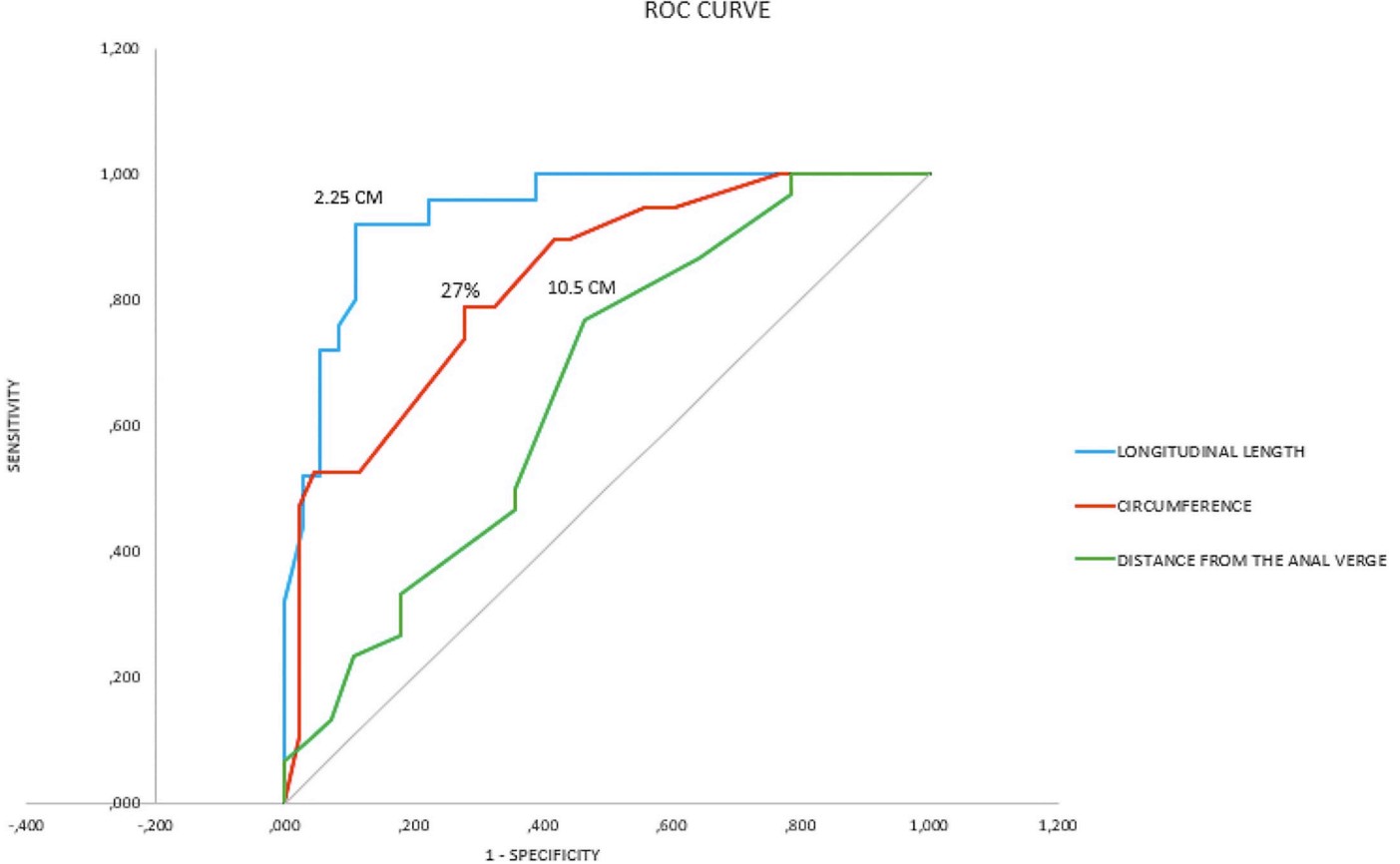

**Fig 7. Graph of ROC curves of the circumference of the affected loop, intestinal nodule size and distance from the anal verge with the respective cutoff points separating linear nodulectomy from segmental resection.**

No intraoperative complications were observed; however, we observed postoperative complications. In the group undergoing linear nodulectomy, one patient (3.7%) developed leakage, which was treated with cavity drainage and antibiotic therapy. Another patient (3.7%) developed stenosis of the anastomosis area, and after failure of dilation attempts, she underwent a new intestinal surgery using the segmental resection technique.

In the group undergoing segmental resection, four patients (11.1%) had complications: in one patient (2.7%), a fistula occurred, which resolved without the need for reoperation; in another, urinary retention occurred, which was resolved with instructions to the patient, urinary catheterization and use of bethanechol; transrectal bleeding followed by leakage was observed in another patient, with no progression to fistula; and another patient exhibited difficulty urinating, which was resolved with physical therapy. There was no significant difference between the groups with respect to complications (Table 1).

## Discussion

Analyzing the distance from the anal verge of the intestinal nodule, we identified a value of 10.5 cm as the cutoff point separating the techniques. Smaller distances were associated with the segmental resection technique, and distances greater than this value were associated with the linear nodulectomy technique. In a study of patients undergoing segmental resection, Malzoni et al., 2016 [29] found distances from the anal verge between 4 and 12 cm, with a distance

smaller or equal to 4 cm in only 6% of cases. Our data shows that the linear nodulectomy technique is useful for upper rectal nodules as demonstrated by our median of 10.5 cm in this group. For lower lesions, the discoid technique may be a better option than linear nodulectomy, as the manipulation of the linear stapler is challenging in the deep pelvis.

We observed that the depth to the muscular and serous layer, on TVUSBP, is more closely associated with the linear nodulectomy technique, whereas the depth to the submucosal and mucosal layers is more closely associated with the segmental resection technique, with statistical significance (p<0.001). Moawad et al., 2011 [30] also showed greater mucosal involvement in 61.5% of patients when the segmental resection technique was performed, compared to 0% of cases when using discoid nodulectomy.

On the ROC curve, a value of 2.25 cm for the length of the intestinal nodule was identified as the cutoff separating the two surgical techniques. Moawad et al., 2011 [30] compared the diameters of intestinal nodules and found a value of 35 mm for the segmental resection technique and 28 mm for discoid resection. Bray-Beraldo et al., 2018 [9] also used a value that corroborates our results, with 30 mm as the parameter differentiating discoid nodulectomy and segmental resection. In a case series undergoing segmental resection, Malzoni et al., 2016 [29] observed that the nodules were no smaller than 3 cm and reached 7 cm. Patients with vaginal endometriotic nodules had significantly larger intestinal nodules (1.5 cm x 3.6 cm; p = 0.019). Although linear nodulectomy excision is ultimately destined to small nodules, because of the risk of stenosis, it seems to us that in this specific condition, it can be a feasible option for discoid nodulectomy.

In the group undergoing the linear nodulectomy technique, our data showed that one patient (3.7%) developed leakage and in the segmental resection group, we reported one patient (2.7%) that a fistula occurred and another patient (2.7%) in which leakage was observed. At the FRIENDS survey [31], the rate of rectovaginal fistula in patients managed by discoid nodulectomy was 3.6%, comparable to 3.9% segmental resection (3.9%). Roman et al. [32], observed a rate of rectovaginal fistula as high as 7.2% for discoid nodulectomy, with high prevalence of this event in patients with low rectovaginal endometriosis—rectal nodules 5.5 cm above the anus. In our data, the TVUSBP for the patient undergoing linear nodulectomy that developed leakage, showed intestinal endometriosis 10cm from the anal verge. And for the patient undergoing segmental resection that developed leakage, the nodule was 12cm from the anal verge.

Regarding the circumference of the loop affected by the intestinal endometriotic nodule, we identified a value of 27% as the cutoff point between the linear nodulectomy and segmental resection techniques. This data corroborates with numeric data, the experts consensus in literature that indicates that nodulectomy should be performed in nodules affecting less than 30% of the circumference of the intestinal loop [24]. Concerning the circumference of the loop affected, there is no high quality cohort study comparing nodulectomy and segmental resection but, some authors suggest that intestinal nodules that infiltrate more than the internal muscular layer or that affect more than 40% of the circumference of the loop should be subjected to segmental resection [13]. This is because nodulectomy of larger nodules may lead to stenosis of the stapling area [28, 33–35]. Meanwhile, the long term follow up study of Mabrouk et al. in 2018 suggests that a conservative approach is prefferd over radical surgery in patients with intermediate risk of bowel segmental resection [36].

A possible limitation of our study would be the evaluation of a microscopic residual lesion after the techniques of linear nodulectomy and segmental resection. The literature shows rates of compromised margin, in cases of intestinal resection, ranging from 10% to 22% of cases [37, 38], and endometriosis microfocuses may be present in 19% of cases, up to 3 cm from the removed lesion. [39]. Studies show, however, that by removing the central focus of

endometriosis, the possible residual microscopic foci are not able to develop [40, 41], therefore, according to this last data, even a more conservative technique, such as linear nodulectomy, would have no restrictions on its use regarding the possibility of recurrence.

In addition, we should highlight the small sample size as a limitation to our study, despite the fact that the pilot project showed statistical significance, with a 99.9% test power with 27 patients. Despite the analysis of the surgery videos and of the medical records, performed retrospectively, it was a strict analysis, using the same criteria for all patients. The gynecologist surgeons HSAAR and PAAR operated together with the digestive tract surgeon FCMR on all patients, so the surgeries were performed with the same surgical team, using the same criteria for choosing one surgical technique or the other.

In this study, we evaluated the surgical techniques more frequently performed in our hospital, segmental resection and linear nodulectomy. The shaving technique although frequently used in other center and commonly seen in literature, is not a common practice in our department. For the future, we are planning to perform a new research that may include shaving, discoid and linear resection techniques.

Considering that we have already demonstrated in previous studies from our group an enhancement of the quality of life [7, 42] after the surgical treatment of intestinal endometriosis with the herein described procedures, we are comfortable to select the surgical technique based on the described criteria of the TVUSBP and confirmed intraoperatively by the surgical team.

In our study, we observed 12 cases of patients with two intestinal injuries described in USTVPI. In 4 cases, it was opted, in the intraoperative, for the linear nodulectomy technique in each lesion individually; in 8 cases, the segmental resection technique was chosen, encompassing both lesions. With only 4 cases in one of the groups, it was not possible to numerically determine, the parameters of choice to perform the intestinal resection technique encompassing both lesions or to perform linear nodulectomy in each nodule. We could hypothesize that this choice is related to the distance between the lesions or to some other variable, such as, for example, the longitudinal length of the lesions, the affected layer, the circumference of the lesions or the distance from the anal border. However, there was a tendency towards greater distance between the nodules in the group undergoing segmental resection.

In literature, there is no defined numerical criteria for the choice of the surgical technique to be used in the treatment of intestinal deep endometriosis. Still, excising the disease by segmental resection or nodulectomy (either shaving, discoid or linear) is based on the surgeon's preference or experience, and several surgeons have published opinions based on their own practical experience [17].

Our pioneering study provides numerical parameters of the intestinal nodule that can be used to guide the choice of which surgical technique to use for resection of the intestinal nodule, such as a nodule diameter of 2.25 cm, distance from the anal verge of 10.5 cm and circumference of the loop affected of 27%. It was not possible to perform a statistical test by analyzing the four nodule parameters (circumference, length, distance from the anal verge and affected layer) together to identify a single cutoff point separating the two techniques. Given the results obtained, in our practice, we consider the circumference of the affected loop and the nodule length as the most important parameters in the preoperative period for guiding the surgeon regarding the surgical technique to be performed. Using only the distance from the anal verge would not be sufficient to determine the most ideal surgical technique.

We found important data in our study regarding the duration of symptoms. In patients diagnosed 2 years prior, we found a smaller extent of intestinal disease, and they were subjected to a less invasive technique to resolve intestinal endometriosis. Patients diagnosed 3

years prior had more extensive disease that required a more invasive surgical technique with segmental resection, which also resulted in longer surgical time.

In this study we treated 45 pacients with drugs before surgery. Considering that the use of combined oral contraceptive in women with posterior infiltrating endometriosis may influence the progression of the nodule size, and symptoms as dismenorrhea and dispareunia [43, 44], a new assessment of these factors had to be made prior to surgery. A new TVUSBP was performed in pacients that had out of date exams, or which the results coud have been modified due to any pre treatment. The results presented and considered for this study were based on the more recent examination available for each patient. Also, the use of drugs has not influenced the choice of surgical technique, once it was defined during the procedure, with visual confirmation of the size, depth and number of lesions.

In our study, we observed that the choice of the surgical technique in the treatment of intestinal endometriosis is not influenced by other variables in TVUSBP, such as the presence of endometrioma, left ovarian mobility, involvement of the left or right uterosacral ligament, retrocervical nodule and ureter involvement.

The data generated by our study is of great importance in the preoperative evaluation of patients to prevent incomplete or suboptimal surgeries due to technical inability of the surgeon, as the surgeon would already be prepared for the degree of surgical difficulty to be expected. Additionally, this data provides a basis for requesting appropriate surgical materials necessary for surgery (staplers, drains, etc.) and, depending on the experience of the pelvic/gynecologist surgeon, a gastrointestinal (GI) surgeon. Another important consideration is the ability to instruct the patient, in the preoperative period, in the surgical technique to be performed, as techniques may differ in the rate of surgical complications, surgical time, morbidity, alteration of bowel habits in the postoperative period and length of hospital stay.

## Conclusion

Transvaginal ultrasonography with bowel preparation for endometriosis mapping was shown to be an effective tool to assist decision-making about the surgical technique to be performed for the treatment of intestinal endometriosis. After obtaining the ROC curve, we determined cutoff values for the longitudinal length of the intestinal nodule (2.25 cm), circumference of the loop (27%) and distance from the anal verge (10.5 cm) separating the segmental resection and linear nodulectomy techniques. Regarding the intestinal layer, we observed that the depth reaching the muscular layer on TVUSBP is more closely associated with the linear nodulectomy technique, while the depth to the submucosal layer is more closely associated with the segmental resection technique.

## Author Contributions

**Conceptualization:** Helizabet Abdalla-Ribeiro, Marina Miyuki Maekawa, Raquel Ferreira Lima, Paulo Ayroza Ribeiro.

**Data curation:** Helizabet Abdalla-Ribeiro, Marina Miyuki Maekawa, Raquel Ferreira Lima, Ana Luisa Alencar de Nicola, Francisco Cesar Martins Rodrigues, Paulo Ayroza Ribeiro.

**Formal analysis:** Helizabet Abdalla-Ribeiro, Marina Miyuki Maekawa, Raquel Ferreira Lima, Ana Luisa Alencar de Nicola, Francisco Cesar Martins Rodrigues, Paulo Ayroza Ribeiro.

**Funding acquisition:** Helizabet Abdalla-Ribeiro, Marina Miyuki Maekawa, Paulo Ayroza Ribeiro.

**Investigation:** Helizabet Abdalla-Ribeiro, Marina Miyuki Maekawa, Francisco Cesar Martins Rodrigues, Paulo Ayroza Ribeiro.

**Methodology:** Helizabet Abdalla-Ribeiro, Marina Miyuki Maekawa, Raquel Ferreira Lima, Ana Luisa Alencar de Nicola, Francisco Cesar Martins Rodrigues, Paulo Ayroza Ribeiro.

**Project administration:** Helizabet Abdalla-Ribeiro, Marina Miyuki Maekawa, Paulo Ayroza Ribeiro.

**Resources:** Helizabet Abdalla-Ribeiro, Marina Miyuki Maekawa, Ana Luisa Alencar de Nicola, Paulo Ayroza Ribeiro.

**Software:** Helizabet Abdalla-Ribeiro, Marina Miyuki Maekawa, Paulo Ayroza Ribeiro.

**Supervision:** Helizabet Abdalla-Ribeiro, Marina Miyuki Maekawa, Paulo Ayroza Ribeiro.

**Validation:** Helizabet Abdalla-Ribeiro, Marina Miyuki Maekawa, Paulo Ayroza Ribeiro.

**Visualization:** Helizabet Abdalla-Ribeiro, Marina Miyuki Maekawa, Ana Luisa Alencar de Nicola, Paulo Ayroza Ribeiro.

**Writing – original draft:** Helizabet Abdalla-Ribeiro, Marina Miyuki Maekawa, Raquel Ferreira Lima, Ana Luisa Alencar de Nicola, Francisco Cesar Martins Rodrigues, Paulo Ayroza Ribeiro.

**Writing – review & editing:** Helizabet Abdalla-Ribeiro, Marina Miyuki Maekawa, Raquel Ferreira Lima, Paulo Ayroza Ribeiro.

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
