## [Decision Letter · Decision Letter 0]

22 Oct 2020

PONE-D-20-26728

Intestinal endometriotic nodules with a length greater than 2.25 cm and affecting more than 27% of the circumference are more likely to undergo segmental resection, rather than nodulectomy.

PLOS ONE

Dear Dr. Maekawa,

Thank you for submitting your manuscript to PLOS ONE. After careful consideration, we feel that it has merit but does not fully meet PLOS ONE’s publication criteria as it currently stands. Therefore, we invite you to submit a revised version of the manuscript that addresses the points raised during the review process.

We look forward to receiving your revised manuscript.

Kind regards,

Diego Raimondo

Academic Editor

PLOS ONE

Journal Requirements:

2.Thank you for including your ethics statement:  "the form consent was not obtained because the data were analyzed retrospectively, without evaluation, contact or interview with the patients. After the surgical procedure patients have no clinical follow up at the institution".   

Please amend your current ethics statement to include the full name of the ethics committee that approved your specific study.

In the ethics statement in the manuscript and in the online submission form, please provide additional information about the patient records used in your retrospective study, including: a) whether all data were fully anonymized before you accessed them; c) the date range (month and year) during which patients whose medical records were selected for this study sought treatment; d) whether the ethics committee waived the need for informed consent for patient records to be used in research. If patients provided informed written consent to have data from their medical records used in research, please include this information."

For additional information about PLOS ONE submissions requirements for ethics oversight of animal work, please refer to http://journals.plos.org/plosone/s/submission-guidelines#loc-animal-research  

3. Please include a copy of Table 1 which you refer to in your text on pages 10,11 and 15.

Reviewers' comments:

Reviewer's Responses to Questions

**Comments to the Author**

1. Is the manuscript technically sound, and do the data support the conclusions?

Reviewer #1: Yes

Reviewer #2: Yes

2. Has the statistical analysis been performed appropriately and rigorously? 

Reviewer #1: Yes

Reviewer #2: Yes

3. Have the authors made all data underlying the findings in their manuscript fully available?

Reviewer #1: Yes

Reviewer #2: Yes

4. Is the manuscript presented in an intelligible fashion and written in standard English?

Reviewer #1: No

Reviewer #2: No

5. Review Comments to the Author

Reviewer #1: “Intestinal endometriotic nodules with a lenght greater than 2.25 cm and affecting more than 27% of the circumference are more likely to undergo segmental resection, rather than nodulectomy”

This retrospective study investigates the efficacy of intestinal ultrasonography with bowel preparation (TVUSBP) for endometriosis mapping in evaluating intestinal endometriosis in order to choose the surgical technique (segmental resection or nodulectomy).

Materials and methods:

- Consent for data collection and data analyses is mandatory, even in retrospective studies.

- Is the surgeon blinded respect to TVUSBP findings? Please provide more explanation about the study design.

- What are inclusions criterias? Please indicate a better explanation of initial inclusions criterias for sample selection.

- Line 146: Please specify how the measurements were performed; please provide references about the US parameters used for nodules characterization.

Results:

- Why did you perform surgery in asymptomatic patients with posterior compartment DIE?

- Line 245: When did you perform the study? In 2019? If so, why the study period indicated in Materials and Methods was from 2010 to 2014?

- Please provide a table for ROC curves parameters including sensitivity, specificity, NPV and PPV for each surgical technique.

- Can we give a look to Table 1? We could not find it in the manuscript.

Discussion:

- Please provide limits of the study (small sample size, retrospective analysis, different surgeons skills).

Additionally, the paper requires major English language revisions.

Reviewer #2: 1- The study design and the sample selection are unclear. Explain inclusion and exclusion criteria.

3- Is the technique (nodulectomy vs resection) chosen by the surgeon during the surgery or is he informed preoperatively about the TVUSBP findings?

4- This is a relevant topic. Parameters to tailor the surgical treatment are crucial but why didn’t you consider the shaving technique in this study? Did you perform surgery in asymptomatic patients? Did you consider the presence of multiple lesions or the bowel nodule only? Mutiple nodules needs a more radical surgery. Please discuss some relevant articles on this topic (example: Mabrouk M, Raimondo D, Altieri M, Arena A, Del Forno S, Moro E, Mattioli G, Iodice R, Seracchioli R. Surgical, Clinical, and Functional Outcomes in Patients with Rectosigmoid Endometriosis in the Gray Zone: 13-Year Long-Term Follow-up. J Minim Invasive Gynecol. 2019 Sep-Oct;26(6):1110-1116. doi: 10.1016/j.jmig.2018.08.031. Epub 2018 Nov 9. PMID: 30414996.)

5- The surgical tecnique is clear. Can you explain better the parameters you used on ultrasound?

6- You have to discuss better the limits of your study:

This is a retrospective analysis of a small population, the linear stapler technique is not commonly used so your results are not generalizable. When you talk about nodulectomy please specify “linear” (also in the title)

7- The paper requires English language revisions.

6. PLOS authors have the option to publish the peer review history of their article (what does this mean?). If published, this will include your full peer review and any attached files.

Reviewer #1: No

Reviewer #2: No

---

## [Author Response · Author response to Decision Letter 0]

8 Jan 2021

Response to Reviewers

We are pleased to resubmit for publication on the revised version of PONE-D-20-26728: “Intestinal endometriotic nodules with a length greater than 2.25 cm and affecting more than 27% of the circumference are more likely to undergo segmental resection, rather than linear nodulectomy”.

We appreciated the constructive criticisms of the Academic Editor and the reviewers. We have addressed each of their concerns as outlined below. The major changes were: we included, as suggested by the both reviewers, more clear and complete description of the criteria for sample selection. Also, as solicited, a major improvement in references for the ultrasonography parameters was made. Finally, following the reviewer’s insightful suggestion, we have provided a broad discussion of the limits of the study, and stressed why it´s findings are still important. In addition, we have rewritten parts of the paper to provide more clarity and corrected the language errors. 

RESPONSE TO THE EDITOR´S COMMENTS

The manuscript was formatted to meet the style requirements.

2. Please provide additional information about the patient records used in your retrospective study, including: a) whether all data were fully anonymized before you accessed them; c) the date range (month and year) during which patients whose medical records were selected for this study sought treatment; d) whether the ethics committee waived the need for informed consent for patient records to be used in research. If patients provided informed written consent to have data from their medical records used in research, please include this information."

We thank the editor for pointing this out. This information is now available at the manuscript as follows:

“The consent form for this specific study was not obtained because in our institution an informed consent is not necessary for retrospective studies without evaluation, contact, questionnaire or interviews with the patients. In addition once admitted at the institution provide written consent to have data of their medical records used in research. All data was fully anonymized before the authors accessed them. The patients selected underwent surgery from Jan/2004 to Dec/2010. One year after the surgical procedure, patients had no clinical follow up at the institution and continue with routine care at primary health care units.”

3. Please include a copy of Table 1 which you refer to in your text on pages 10,11 and 15.

We apologize for the missing table. It was included in this new version.

RESPONSE TO REVIEWER #1 COMMENTS:

1- Consent for data collection and data analyses is mandatory, even in retrospective studies.

We agree and have updated this part. A new paragraph was included in the manuscript, concerning the use of patient medical records for scientific studies, as follows:

“The consent form for this specific study was not obtained because in our institution an informed consent is not necessary for retrospective studies without evaluation, contact, questionnaire or interviews with the patients. In addition, once admitted at the institution the patient provides written consent to have data of their medical records used in research. All data was fully anonymized before the authors accessed them. The patients selected, underwent surgery from Jan/2004 to Dec/2010. One year after the surgical procedure, patients had no clinical follow up at the institution and continue with routine care at primary health care units.”

2- Is the surgeon blinded respect to TVUSBP findings? Please provide more explanation about the study design.

We thank the reviewer for the observation. A new explanation of this part of the study´s design was included: 

“… They were submitted to ultrasonography, performed by a single examiner, suggesting intestinal endometriotic nodules - or images that the examiner can presume intestinal adhesion is present. All patients underwent videolaparoscopy surgery for intestinal endometriosis resection with endometriosis confirmation in anatomopathological examination. The surgeons were aware of the USTVBP result.” 

3- What are inclusion criteria? Please indicate a better explanation of initial inclusion criteria for sample selection.

We recognize that this part needed further elaboration. The following part was added for better explanation of the sample selection:

“We included in this study all patients submitted to surgical treatment of bowel endometriosis in our department, by the same surgical team (P.A.A.R., H.S.A.A.R. and F.C.M.R.), during the specified period (2010-2014) regardless of age, parity, previous hormonal or surgical treatment for endometriosis or associated procedures. All patients had a TVUSBP examination performed by a single radiologist (A.L.A.N.) and their surgeries were recorded on DVD. Histological confirmation of intestinal endometriosis was a mandatory inclusion criteria.” 

4- Line 146: Please specify how the measurements were performed; please provide references about the US parameters used for nodule characterization.

This part was rewritten including more references and pictures for better explanation:

“Transvaginal ultrasonography with bowel preparation for endometriosis mapping (USTVBP) was performed according to the protocol of our department (19) and literature (23, 27, 28). All examinations […]transducers.”

“Intestinal deep infiltrating endometriosis lesions were defined as hypoechoic nodular thickening with regular or toothed margins (comet shape) or hypoechoic linear thickening with regular or irregular margins and involvement of the muscular or submucosa layers (26, 28).”

The studies cited here are:

19. Lima R, Abdalla-Ribeiro H, Nicola AL, Eras A, Lobao A, Ribeiro PA. Endometriosis on the uterosacral ligament: a marker of ureteral involvement. Fertility and Sterility 2017; 107(6), 1348–1354. 

23. Cardoso MM, Junior HW, Berardo PT, Junior AC, Domingues MN, Gasparetto EL, et al. Evaluation of agreement between transvaginal ultrasonography and magnetic resonance imaging of the pelvis in deep endometriosis with emphasis on intestinal involvement. Radiol Bras 2009;42:89–95.

26. Reid S, Lu C, Hardy N, Casikar I, Reid G, Cario G, et al. Office gel sonovaginography for the prediction of posterior deep infiltrating endometriosis: a multicenter prospective observational study. Ultrasound Obstet Gynecol. 2014; 44(6):710-8.

27. Abrão MS, Gonçalves MO, Dias JA Jr, Podgaec S, Chamie LP, Blasbalg R. Comparison between clinical examination, transvaginal sonography and magnetic resonance imaging for the diagnosis of deep endometriosis. Hum Reprod. 2007; 22(12):3092-7.

28. Goncalves MO, Podgaec S, Dias JA Jr, Gonzalez M, Abrao MS. Transvaginal ultrasonography with bowel preparation is able to predict the number of lesions and rectosigmoid layers affected in cases of deep endometriosis, defining surgical strategy. Hum Reprod 2010;25:665–71.

5- Why did you perform surgery in asymptomatic patients with posterior compartment DIE?

We thank the reviewer for pointing this out, so the following information was added to the manuscript: “These patients had clinical symptoms, complaints of infertility or physical examinations that would suggest endometriosis.”

6- Line 245: When did you perform the study? In 2019? If so, why was the study period indicated in Materials and Methods from 2010 to 2014?

We apologize for the confusing information about the dates. The following paragraph was rewritten to clarify this issue:

“In 2019 a cross-sectional observational study was performed and submitted to statistical analysis. This study included 111 medical records of patients diagnosed with endometriosis who underwent videolaparoscopy surgery from April 2010 to November 2014.”

7- Please provide a table for ROC curve parameters including sensitivity, specificity, NPV and PPV for each surgical technique.

We agree and added the tables to the manuscript.

8- Can we take a look at Table 1? We could not find it in the manuscript.

We apologize for the missing table. It was included in this new version.

9- Please provide limits of the study (small sample size, retrospective analysis, different surgeons skills).

We appreciate the reviewer’s suggestion, and added further discussion about the limits of the study, as follows:

“In addition, we should highlight the small sample size as a limitation to our study, despite the fact that the pilot project showed statistical significance, with a 99.9% test power with 27 patients. Despite the analysis of the surgery videos and of the medical records, performed retrospectively, it was a strict analysis, using the same criteria for all patients. The gynecologist surgeons HSAAR and PAAR operated together with the digestive tract surgeon FCMR on all patients, so the surgeries were performed with the same surgical team, using the same criteria for choosing one surgical technique or the other.” 

10- Additionally, the paper requires major English language revisions.

The paper was rewritten to improve clarity and correct mistakes concerning the English language. We hope it’s now easier to understand.

RESPONSE TO REVIEWER #2 COMMENTS:

1- The study design and the sample selection are unclear. Explain inclusion and exclusion criteria.

We recognize that this part needed further elaboration. The following part was added for better explanation of the sample selection:

 “We included in this study all patients submitted to surgical treatment of bowel endometriosis in our department, by the same surgical team (P.A.A.R., H.S.A.A.R. and F.C.M.R.), during the specified period (2010-2014) regardless of age, parity, previous hormonal or surgical treatment for endometriosis or associated procedures. All patients had a TVUSBP examination performed by a single radiologist (A.L.A.N.) and their surgeries were recorded on DVD. Histological confirmation of intestinal endometriosis was a mandatory inclusion criteria.” 

“The exclusion criteria for the study included: loss of medical records, preoperative diagnosis of endometriosis by another image exam, preoperative TVUSBP performed by another radiologist, intestinal resection not performed, intestinal resection done without the use of stapling; surgeries performed by a different surgical team.”

2- Is the technique (nodulectomy vs resection) chosen by the surgeon during the surgery or is he informed preoperatively about the TVUSBP findings?

We thank the reviewer for pointing this out. This part is now better explained at the study design section:

“The surgeries were performed by senior surgeons with extensive experience in the treatment of deep infiltrating endometriosis (P.A.A.R., H.S.A.A.R. and F.C.M.R.). The surgeons were aware of the USTVBP result.”

“Although the USTVBP can define preoperatively the dimensions of the lesions and suggest one technique or another, the final decision on the surgical technique to be performed (linear nodulectomy or segmental resection) was established intraoperatively, by performing a rectal lumen diameter test, with the insertion of a 29 mm diameter rectal probe. If lumen stenosis was observed, the segmental resection technique was chosen.”

3- This is a relevant topic. Parameters to tailor the surgical treatment are crucial but why didn’t you consider the shaving technique in this study? Did you perform surgery in asymptomatic patients? Did you consider the presence of multiple lesions or the bowel nodule only? Multiple nodules needs a more radical surgery. Please discuss some relevant articles on this topic.

We have elaborated some new explanations about all these issues. We hope the paper is clearer now.

About the shaving technique we made the following explanation: “In this study, we evaluated the surgical techniques more frequently performed in our hospital, segmental resection and linear nodulectomy. The shaving technique although frequently used in other centers and commonly seen in literature, is not a common practice in our department. In the future, we are planning to perform a new research that may include shaving, discoid and linear resection techniques.”

About the patients that didn´t have clinical symptoms, they were submitted to surgery due to infertility complaints.

Finally, about the multiple lesions issue, we thank the reviewer for this insightful suggestion, and added the following discussion:

“ In our study, we observed that the choice of surgical technique in the treatment of intestinal endometriosis is not influenced by other variables in TVUSBP, such as the presence of endometrioma, left ovarian mobility, involvement of the left or right uterosacral ligament, retrocervical nodule and ureter involvement.”

“…This is because nodulectomy of larger nodules may lead to stenosis of the stapling area (28, 33-35). Meanwhile, the long term follow up study of Mabrouk et al. in 2018 suggests that a conservative approach is preferred over radical surgery in patients with intermediate risk of bowel segmental resection (36).”

The paper cited here is:

36. Mabrouk M, Raimondo D, Altieri M, Arena A, Del Forno S, Moro E, Mattioli G, Iodice R, Seracchioli R. Surgical, Clinical, and Functional Outcomes in Patients with Rectosigmoid Endometriosis in the Gray Zone: 13-Year Long-Term Follow-up. J Minim Invasive Gynecol. 2019 Sep-Oct;26(6):1110-1116. doi: 10.1016/j.jmig.2018.08.031. Epub 2018 Nov 9. PMID: 30414996.)

4- The surgical technique is clear. Can you better explain the parameters you used on ultrasound?

We agree and this part was rewritten including more references and pictures for better explanation:

“Transvaginal ultrasonography with bowel preparation for endometriosis mapping (USTVBP) was performed according to the protocol of our department (19) and literature (23, 27, 28). All examinations […]transducers.”

“Intestinal deep infiltrating endometriosis lesions were defined as hypoechoic nodular thickening with regular or toothed margins (comet shape) or hypoechoic linear thickening with regular or irregular margins and involvement of the muscular or submucosa layers (26, 28).”

The studies cited here are:

19. Lima R, Abdalla-Ribeiro H, Nicola AL, Eras A, Lobao A, Ribeiro PA. Endometriosis on the uterosacral ligament: a marker of ureteral involvement. Fertility and Sterility 2017; 107(6), 1348–1354. 

23. Cardoso MM, Junior HW, Berardo PT, Junior AC, Domingues MN, Gasparetto EL, et al. Evaluation of agreement between transvaginal ultrasonography and magnetic resonance imaging of the pelvis in deep endometriosis with emphasis on intestinal involvement. Radiol Bras 2009;42:89–95.

26. Reid S, Lu C, Hardy N, Casikar I, Reid G, Cario G, et al. Office gel sonovaginography for the prediction of posterior deep infiltrating endometriosis: a multicenter prospective observational study. Ultrasound Obstet Gynecol. 2014; 44(6):710-8.

27. Abrão MS, Gonçalves MO, Dias JA Jr, Podgaec S, Chamie LP, Blasbalg R. Comparison between clinical examination, transvaginal sonography and magnetic resonance imaging for the diagnosis of deep endometriosis. Hum Reprod. 2007; 22(12):3092-7.

28. Goncalves MO, Podgaec S, Dias JA Jr, Gonzalez M, Abrao MS. Transvaginal ultrasonography with bowel preparation is able to predict the number of lesions and rectosigmoid layers affected in cases of deep endometriosis, defining surgical strategy. Hum Reprod 2010;25:665–71.

5- You have to better discuss the limits of your study: This is a retrospective analysis of a small population, the linear stapler technique is not commonly used so your results are not generalizable.

We appreciate reviewer´s suggestion, and added further discussion about the limits of the study, as follows:

 “In addition, we should highlight the small sample size as a limitation to our study, despite the fact that the pilot project showed statistical significance, with a 99.9% test power with 27 patients. Despite the analysis of the surgery videos and of the medical records, performed retrospectively, it was a strict analysis, using the same criteria for all patients. The gynecologist surgeons HSAAR and PAAR operated together with the digestive tract surgeon FCMR on all patients, so the surgeries were performed with the same surgical team, using the same criteria for choosing one surgical technique or the other.”

“In this study, we evaluated the surgical techniques more frequently performed in our hospital, segmental resection and linear nodulectomy. The shaving technique although frequently used in other centers and commonly seen in literature, is not a common practice in our department. For the future, we are planning to perform a new research that may include shaving, discoid and linear resection techniques.”

“Considering that we have already demonstrated in previous studies from our group an enhancement of the quality of life after the surgical treatment of intestinal endometriosis with the herein described procedures, we are comfortable to select the surgical technique based on the described criteria of the TVUSBP and confirmed intraoperatively by the surgical team. “

6- When you talk about nodulectomy please specify “linear” (also in the title)

The word “linear” was included prior to nodulectomy all throughout the paper.

7- The paper requires English language revisions.

The paper was rewritten to improve clarity and correct mistakes concerning the English language. We hope it´s now easier to understand.

Once again we would like to thank the reviewers for taking the time to analyze our manuscript. We look forward to hearing from you regarding our submission, and to respond to any further questions you may have.

Sincerely,

Marina M. Maekawa, M.D.

December 21th, 2020

---

## [Decision Letter · Decision Letter 1]

20 Jan 2021

PONE-D-20-26728R1

Intestinal endometriotic nodules with a length greater than 2.25 cm and affecting more than 27% of the circumference are more likely to undergo segmental resection, rather than linear nodulectomy.

PLOS ONE

Dear Dr. Maekawa,

Thank you for submitting your manuscript to PLOS ONE. After careful consideration, we feel that it has merit but does not fully meet PLOS ONE’s publication criteria as it currently stands. Therefore, we invite you to submit a revised version of the manuscript that addresses the points raised during the review process.

We look forward to receiving your revised manuscript.

Kind regards,

Diego Raimondo

Academic Editor

PLOS ONE

Reviewers' comments:

Reviewer's Responses to Questions

**Comments to the Author**

1. If the authors have adequately addressed your comments raised in a previous round of review and you feel that this manuscript is now acceptable for publication, you may indicate that here to bypass the “Comments to the Author” section, enter your conflict of interest statement in the “Confidential to Editor” section, and submit your "Accept" recommendation.

Reviewer #2: All comments have been addressed

2. Is the manuscript technically sound, and do the data support the conclusions?

Reviewer #2: Yes

3. Has the statistical analysis been performed appropriately and rigorously? 

Reviewer #2: Yes

4. Have the authors made all data underlying the findings in their manuscript fully available?

Reviewer #2: Yes

5. Is the manuscript presented in an intelligible fashion and written in standard English?

Reviewer #2: Yes

6. Review Comments to the Author

Reviewer #2: just one more comment:

You treated 18 patients in the linear nodulectomy group and 27 patients in the segmental resection group (table 1) with drugs before surgery. Did the previous treatment have some influence on the nodule and/or on the surgical choice? Please comment your evidences with the literature (example PMID: 21777836)

7. PLOS authors have the option to publish the peer review history of their article (what does this mean?). If published, this will include your full peer review and any attached files.

Reviewer #2: No

---

## [Author Response · Author response to Decision Letter 1]

9 Feb 2021

Dear Editor:

We are pleased to resubmit for publication on the revised version of PONE-D-20-26728: “Intestinal endometriotic nodules with a length greater than 2.25 cm and affecting more than 27% of the circumference are more likely to undergo segmental resection, rather than linear nodulectomy”.

We appreciated the constructive criticisms of the Academic Editor and the reviewers. We have addressed each of their concerns as outlined below. The major changes were: we included, as suggested by the both reviewers, more clear and complete description of the criteria for sample selection. Also, as solicited, a major improvement in references for the ultrasonography parameters was made. Finally, following the reviewer’s insightful suggestion, we have provided a broad discussion of the limits of the study, and stressed why it´s findings are still important. In addition, we have rewritten parts of the paper to provide more clarity and corrected the language errors.

---

## [Editor Report · Decision Letter 2]

11 Feb 2021

Intestinal endometriotic nodules with a length greater than 2.25 cm and affecting more than 27% of the circumference are more likely to undergo segmental resection, rather than linear nodulectomy.

PONE-D-20-26728R2

Dear Dr. Maekawa,

We’re pleased to inform you that your manuscript has been judged scientifically suitable for publication and will be formally accepted for publication once it meets all outstanding technical requirements.

Kind regards,

Diego Raimondo

Academic Editor

PLOS ONE

---

## [Editor Report · Acceptance letter]

4 Mar 2021

PONE-D-20-26728R2 

Intestinal endometriotic nodules with a length greater than 2.25 cm and affecting more than 27% of the circumference are more likely to undergo segmental resection, rather than linear nodulectomy. 

Dear Dr. Maekawa:

I'm pleased to inform you that your manuscript has been deemed suitable for publication in PLOS ONE. Congratulations! Your manuscript is now with our production department. 

Kind regards, 

on behalf of

Dr. Diego Raimondo 

Academic Editor

PLOS ONE